# Deep Learning is Singular, and That's Good

## Abstract

In singular models, the optimal set of parameters forms an analytic set with singularities and classical statistical inference cannot be applied to such models. This is significant for deep learning as neural networks are singular and thus "dividing" by the determinant of the Hessian or employing the Laplace approximation are not appropriate. Despite its potential for addressing fundamental issues in deep learning, singular learning theory appears to have made little inroads into the developing canon of deep learning theory. Via a mix of theory and experiment, we present an invitation to singular learning theory as a vehicle for understanding deep learning and suggest important future work to make singular learning theory directly applicable to how deep learning is performed in practice.

## 1 Introduction

It has been understood for close to twenty years that neural networks are singular statistical models (Amari et al., 2003; Watanabe, 2007). This means, in particular, that the set of network weights equivalent to the true model under the Kullback-Leibler divergence forms a real analytic variety which fails to be an analytic manifold due to the presence of singularities. It has been shown by Sumio Watanabe that the geometry of these singularities controls quantities of interest in statistical learning theory, e.g., the generalisation error. Singular learning theory (Watanabe, 2009) is the study of singular models and requires very different tools from the study of regular statistical models. The breadth of knowledge demanded by singular learning theory – Bayesian statistics, empirical processes and algebraic geometry – is rewarded with profound and surprising results which reveal that singular models are different from regular models in practically important ways. To illustrate the relevance of singular learning theory to deep learning, each section of this paper illustrates a key takeaway idea[1].

**The real log canonical threshold (RLCT) is the correct way to count the effective number of parameters in a deep neural network (DNN)** (Section 4). To every (model, truth, prior) triplet is associated a birational invariant known as the real log canonical threshold. The RLCT can be understood in simple cases as half the number of normal directions to the set of true parameters. We will explain why this matters more than the curvature of those directions (as measured for example by eigenvalues of the Hessian) laying bare some of the confusion over "flat" minima.

**For singular models, the Bayes predictive distribution is superior to MAP and MLE** (Section 5). In regular statistical models, the 1) Bayes predictive distribution, 2) maximum a posteriori (MAP) estimator, and 3) maximum likelihood estimator (MLE) have asymptotically equivalent generalisation error (as measured by the Kullback-Leibler divergence). This is not so in singular models. We illustrate in our experiments that even "being Bayesian" in just the final layers improves generalisation over MAP. Our experiments further confirm that the Laplace approximation of the predictive distribution Smith & Le (2017); Zhang et al. (2018) is not only theoretically inappropriate but performs poorly.

**Simpler true distribution means lower RLCT** (Section 6). In singular models the RLCT depends on the (model, truth, prior) triplet whereas in regular models it depends only on the (model, prior) pair. The RLCT increases as the complexity of the true distribution relative to the supposed model increases. We verify this experimentally with a simple family of ReLU and SiLU networks.

---

[1] The code to reproduce all experiments in the paper will be released on Github. For now, see the zip file.

## 2 RELATED WORK

In classical learning theory, generalisation is explained by measures of capacity such as the $l_2$ norm, Radamacher complexity, and VC dimension (Bousquet et al., 2003). It has become clear however that these measures cannot capture the empirical success of DNNs (Zhang et al., 2017). For instance, over-parameterised neural networks can easily fit random labels (Zhang et al., 2017; Du et al., 2018; Allen-Zhu et al., 2019b) indicating that complexity measures such as Rademacher complexity are very large. There is also a slate of work on generalisation bounds in deep learning. Uniform convergence bounds (Neyshabur et al., 2015; Bartlett et al., 2017; Neyshabur & Li, 2019; Arora et al., 2018) usually cannot provide non-vacuous bounds. Data-dependent bounds (Brutzkus et al., 2018; Li & Liang, 2018; Allen-Zhu et al., 2019a) consider the "classifiability" of the data distribution in generalisation analysis of neural networks. Algorithm-dependent bounds (Daniely, 2017; Arora et al., 2019; Yehudai & Shamir, 2019; Cao & Gu, 2019) consider the relation of Gaussian initialisation and the training dynamics of (stochastic) gradient descent to kernel methods (Jacot et al., 2018).

In contrast to many of the aforementioned works, we are interested in estimating the conditional *distribution* $q(y|x)$. Specifically, we measure the generalisation error of some estimate $\hat{q}_n(y|x)$ in terms of the Kullback-Leibler divergence between $q$ and $\hat{q}_n$, see (8). The next section gives a crash course on singular learning theory. The rest of the paper illustrates the key ideas listed in the introduction. Since we cover much ground in this short note, we will review other relevant work along the way, in particular literature on "flatness", the Laplace approximation in deep learning, etc.

## 3 SINGULAR LEARNING THEORY

To understand why classical measures of capacity fail to say anything meaningful about DNNs, it is important to distinguish between two different types of statistical models. Recall we are interested in estimating the true (and unknown) conditional distribution $q(y|x)$ with a class of models $\{p(y|x,w) : w \in W\}$ where $W \subset \mathbb{R}^d$ is the parameter space. We say the model is *identifiable* if the mapping $w \mapsto p(y|x,w)$ is one-to-one. Let $q(x)$ be the distribution of $x$. The Fisher information matrix associated with the model $\{p(y|x,w) : w \in W\}$ is the matrix-valued function on $W$ defined by

$$I(w)_{ij} = \int\int \frac{\partial}{\partial w_i}[\log p(y|x,w)] \frac{\partial}{\partial w_j}[\log p(y|x,w)] q(y|x)q(x)dxdy,$$

if this integral is finite. Following the conventions in Watanabe (2009), we have the following bifurcation of statistical models. A statistical model $p(y|x,w)$ is called **regular** if it is 1) identifiable and 2) has positive-definite Fisher information matrix. A statistical model is called **strictly singular** if it is not regular.

Let $\varphi(w)$ be a prior on the model parameters $w$. To every (model, truth, prior) triplet, we can associate the zeta function, $\zeta(z) = \int K(w)^z \varphi(w)\,dw, z \in \mathbb{C}$, where $K(w)$ is the Kullback-Leibler (KL) divergence between the model $p(y|x,w)$ and the true distribution $q(y|x)$:

$$K(w) := \int\int q(y|x) \log \frac{q(y|x)}{p(y|x,w)} q(x)\,dx\,dy. \tag{1}$$

For a (model, truth, prior) triplet $(p(y|x,w), q(y|x), \varphi)$, let $-\lambda$ be the maximum pole of the corresponding zeta function. We call $\lambda$ the **real log canonical threshold** (RLCT) (Watanabe, 2009) of the (model, truth, prior) triplet. The RLCT is the central quantity of singular learning theory.

By Watanabe (2009, Theorem 6.4) the RLCT is equal to $d/2$ in regular statistical models and bounded above by $d/2$ in strictly singular models if *realisability* holds: let

$$W_0 = \{w \in W : p(y|x,w) = q(y|x)\}$$

be the set of true parameters, we say $q(y|x)$ is **realisable** by the model class if $W_0$ is non-empty. The condition of realisability is critical to standard results in singular learning theory. Modifications to the theory are needed in the case that $q(y|x)$ is not realisable, see the condition called relatively finite variance in Watanabe (2018).

**Neural networks in singular learning theory.** Let $W \subseteq \mathbb{R}^d$ be the space of weights of a neural network of some fixed architecture, and let $f(x,w) : \mathbb{R}^N \times W \longrightarrow \mathbb{R}^M$ be the associated function.

We shall focus on the regression task and study the model

$$p(y|x, w) = \frac{1}{(2\pi)^{M/2}} \exp\left(-\frac{1}{2}\|y - f(x, w)\|^2\right) \tag{2}$$

but singular learning theory can also apply to classification, for instance. It is routine to check (see Appendix A.1) that for feedforward ReLU networks not only is the model strictly singular but the matrix $I(w)$ is degenerate for all nontrivial weight vectors and the Hessian of $K(w)$ is degenerate at every point of $W_0$.

**RLCT plays an important role in model selection.** One of the most accessible results in singular learning theory is the work related to the widely-applicable Bayesian information criterion (WBIC) Watanabe (2013), which we briefly review here for completeness. Let $\mathcal{D}_n = \{(x_i, y_i)\}_{i=1}^n$ be a dataset of input-output pairs. Let $L_n(w)$ be the negative log likelihood

$$L_n(w) = -\frac{1}{n} \sum_{i=1}^n \log p(y_i|x_i, w) \tag{3}$$

and $p(\mathcal{D}_n|w) = \exp(-nL_n(w))$. The marginal likelihood of a model $\{p(y|x, w) : w \in W\}$ is given by $p(\mathcal{D}_n) = \int_W p(\mathcal{D}_n|w)\varphi(w)\, dw$ and can be loosely interpreted as the evidence for the model. Between two models, we should prefer the one with higher model evidence. However, since the marginal likelihood is an intractable integral over the parameter space of the model, one needs to consider some approximation.

The well-known Bayesian Information Criterion (BIC) derives from an asymptotic approximation of $-\log p(\mathcal{D}_n)$ using the Laplace approximation, leading to $\text{BIC} = nL_n(w_{\text{MLE}}) + \frac{d}{2} \log n$. Since we want the marginal likelihood of the data for some given model to be high one should almost never adopt a DNN according to the BIC, since in such models $d$ may be very large. However, this argument contains a serious mathematical error: the Laplace approximation used to derive BIC only applies to *regular* statistical models, and DNNs are not regular. The correct criterion for both regular and strictly singular models was shown in Watanabe (2013) to be $nL_n(w_0) + \lambda \log n$ where $w_0 \in W_0$ and $\lambda$ is the RLCT. Since DNNs are highly singular $\lambda$ may be much smaller than $d/2$ (Section 6) it is possible for DNNs to have high marginal likelihood – consistent with their empirical success.

## 4    VOLUME DIMENSION, EFFECTIVE DEGREES OF FREEDOM, AND FLATNESS

**Volume codimension**. The easiest way to understand the RLCT is as a volume codimension (Watanabe, 2009, Theorem 7.1). Suppose that $W \subseteq \mathbb{R}^d$ and $W_0$ is nonempty, i.e., the true distribution is realisable. We consider a special case in which the KL divergence in a neighborhood of every point $v_0 \in W_0$ has an expression in local coordinates of the form

$$K(w) = \sum_{i=1}^{d'} c_i w_i^2, \tag{4}$$

where the coefficients $c_1, \ldots, c_{d'} > 0$ may depend on $v_0$ and $d'$ may be strictly less than $d$. If the model is regular then this is true with $d = d'$ and if it holds for $d' < d$ then we say that the pair $(p(y|x, w), q(y|x))$ is *minimally singular*. It follows that the set $W_0 \subseteq W$ of true parameters is a regular submanifold of codimension $d'$ (that is, $W_0$ is a manifold of dimension $d - d'$ where $W$ has dimension $d$). Under this hypothesis there are, near each true parameter $v_0 \in W_0$, exactly $d - d'$ directions in which $v_0$ can be varied without changing the model $p(y|x, w)$ and $d'$ directions in which varying the parameters does change the model. In this sense, there are $d'$ *effective parameters* near $v_0$.

This number of effective parameters can be computed by an integral. Consider the volume of the set of almost true parameters $V(t, v_0) = \int_{K(w) < t} \varphi(w) dw$ where the integral is restricted to a small closed ball around $v_0$. As long as the prior $\varphi(w)$ is non-zero on $W_0$ it does not affect the relevant features of the volume, so we may assume $\varphi$ is constant on the region of integration in the first $d'$ directions and normal in the remaining directions, so up to a constant depending only on $d'$ we have

$$V(t, v_0) \propto \frac{t^{d'/2}}{\sqrt{c_1 \cdots c_{d'}}} \tag{5}$$

and we can extract the exponent of $t$ in this volume in the limit

$$d' = 2 \lim_{t \to 0} \frac{\log \left\{ V(at, v_0)/V(t, v_0) \right\}}{\log(a)} \tag{6}$$

for any $a > 0$, $a \neq 1$. We refer to the right hand side of (6) as the *volume codimension* at $v_0$.

The function $K(w)$ has the special form (4) locally with $d' = d$ if the statistical model is regular (and realisable) and with $d' < d$ in some singular models such as reduced rank regression (Appendix A.2). While such a local form does not exist for a singular model generally (in particular for neural networks) nonetheless under natural conditions (Watanabe, 2009, Theorem 7.1) we have $V(t, v_0) = ct^\lambda + o(t^\lambda)$ where $c$ is a constant. We assume that in a sufficiently small neighborhood of $v_0$ the point RLCT $\lambda$ at $v_0$ (Watanabe, 2009, Definition 2.7) is less than or equal to the RLCT at every point in the neighborhood so that the multiplicity $m = 1$, see Section 7.6 of (Watanabe, 2009) for relevant discussion. It follows that the limit on the right hand side of (6) exists and is equal to $\lambda$. In particular $\lambda = d'/2$ in the minimally singular case.

Note that for strictly singular models such as DNNs $2\lambda$ may not be an integer. This may be disconcerting but the connection between the RLCT, generalisation error and volume dimension strongly suggests that $2\lambda$ is nonetheless the only geometrically meaningful "count" of the effective number of parameters near $v_0$.

**RLCT and likelihood vs temperature**. Again working with the model in (2), consider the expectation over the posterior at temperature $T$ as defined in (17) of the negative log likelihood (3)

$$E(T) = \mathbb{E}_w^{1/T} \left[ nL_n(w) \right] = \mathbb{E}_w^{1/T} \left[ \tfrac{1}{2} \sum_{i=1}^n \|y_i - f(x_i, w)\|^2 \right] + \frac{nM}{2} \log(2\pi) \,.$$

Note that when $n$ is large $L_n(v_0) \approx \frac{M}{2} \log(2\pi)$ for any $v_0 \in W_0$ so for $T \approx 0$ the posterior concentrates around the set $W_0$ of true parameters and $E(T) \approx \frac{nM}{2} \log(2\pi)$. Consider the increase $\Delta E = E(T + \Delta T) - E(T)$ corresponding to an increase in temperature $\Delta T$. It can be shown that $\Delta E \approx \lambda \Delta T$ where the reader should see (Watanabe, 2013, Corollary 3) for a precise statement. As the temperature increases, samples taken from the tempered posterior are more distant from $W_0$ and the error $E$ will increase. If $\lambda$ is smaller then for a given increase in temperature the quantity $E$ increases less: this is one way to understand intuitively why a model with smaller RLCT generalises better from the dataset $D_n$ to the true distribution.

**Flatness**. It is folklore in the deep learning community that flatness of minima is related to generalisation (Hinton & Van Camp, 1993; Hochreiter & Schmidhuber, 1997) and this claim has been revisited in recent years (Chaudhari et al., 2017; Smith & Le, 2017; Jastrzebski et al., 2017; Zhang et al., 2018). In regular models this can be justified using the lower order terms of the asymptotic expansion of the Bayes free energy (Balasubramanian, 1997, §3.1) but the argument breaks down in strictly singular models, since for example the Laplace approximation of Zhang et al. (2018) is invalid. The point can be understood via an analysis of the version of the idea in (Hochreiter & Schmidhuber, 1997). Their measure of entropy compares the volume of the set of parameters with tolerable error $t_0$ (our almost true parameters) to a standard volume

$$-\log \left[ \frac{V(t_0, v_0)}{t_0^{d/2}} \right] = \frac{d - d'}{2} \log(t_0) + \tfrac{1}{2} \sum_{i=1}^d \log c_i \,. \tag{7}$$

Hence in the case $d = d'$ the quantity $-\frac{1}{2} \sum_i \log(c_i)$ is a measure of the entropy of the set of true parameters near $w_0$, a point made for example in Zhang et al. (2018). However when $d' < d$ this conception of entropy is inappropriate because of the $d - d'$ directions in which $K(w)$ is flat near $v_0$, which introduce the $t_0$ dependence in (7).

## 5 GENERALISATION

The generalisation puzzle (Poggio et al., 2018) is one of the central mysteries of deep learning. Theoretical investigations into the matter is an active area of research Neyshabur et al. (2017). Many of the recent proposals of capacity measures for neural networks are based on the eigenspectrum of the (degenerate) Hessian, e.g., Thomas et al. (2019); Maddox et al. (2020). But this is not appropriate for singular models, and hence for DNNs.

Since we are interested in learning the *distribution*, our notion of generalisation is slightly different, being measured by the KL divergence. Precise statements regarding the generalisation behavior in singular models can be made using singular learning theory. Let the network weights be denoted $\theta$ rather than $w$ for reasons that will become clear. Recall in the Bayesian paradigm, prediction proceeds via the so-called Bayes predictive distribution, $p(y|x, \mathcal{D}_n) = \int p(y|x, \theta)p(\theta|\mathcal{D}_n)\, d\theta$. More commonly encountered in deep learning practice are the MAP and MLE point estimators. While in a regular statistical model, the three estimators 1) Bayes predictive distribution, 2) MAP, and 3) MLE have the *same* leading term in their asymptotic generalisation behavior, the same is not true in singular models. More precisely, let $\hat{q}_n(y|x)$ be some estimate of the true unknown conditional density $q(y|x)$ based on the dataset $\mathcal{D}_n$. The generalisation error of the predictor $\hat{q}_n(y|x)$ is

$$G(n) := KL(q(y|x)||\hat{q}_n(y|x)) = \int\int q(y|x)\log\frac{q(y|x)}{\hat{q}_n(y|x)}q(x)\, dy\, dx. \tag{8}$$

To account for sampling variability, we will work with the *average generalisation error*, $\mathbb{E}_n G(n)$, where $\mathbb{E}_n$ denotes expectation over the dataset $\mathcal{D}_n$. By Watanabe (2009, Theorem 1.2 and Theorem 7.2), we have

$$\mathbb{E}_n G(n) = \lambda/n + o(1/n) \text{ if } \hat{q}_n \text{ is the Bayes predictive distribution}, \tag{9}$$

where $\lambda$ is the RLCT corresponding to the triplet $(p(y|x, \theta), q(y|x), \varphi(\theta))$. In contrast, we should note that Zhang et al. (2018) and Smith & Le (2017) rely on the Laplace approximation to explain the generalisation of the Bayes predictive distribution though both works acknowledge the Laplace approximate is inappropriate. For completeness, a quick sketch of the derivation of (9) is provided in Appendix A.4. Now by (Watanabe, 2009, Theorem 6.4) we have

$$\mathbb{E}_n G(n) = C/n + o(1/n) \text{ if } \hat{q}_n \text{ is the MAP or MLE}, \tag{10}$$

where $C$ (different for MAP and MLE) is the maximum of some Gaussian process. For regular models, the MAP, MLE, and the Bayes predictive distribution have the same leading term for $\mathbb{E}_n G(n)$ since $\lambda = C = d/2$. However in singular models, $C$ is generally greater than $\lambda$, meaning we should prefer the Bayes predictive distribution for singular models.

That the RLCT has such a simple relationship to the Bayesian generalisation error is remarkable. On the other hand, the practical implications of (19) are limited since the Bayes predictive distribution is intractable. While approximations to the Bayesian predictive distribution, say via variational inference, might inherit a similar relationship between generalisation and the (variational) RLCT, serious theoretical developments will be required to rigorously establish this. The challenge comes from the fact that for approximate Bayesian predictive distributions, the free energy and generalisation error may have different learning coefficients $\lambda$. This was well documented in the case of a neural network with one hidden layer (Nakajima & Watanabe, 2007).

We set out to investigate whether certain very simple approximations of the Bayes predictive distribution can already demonstrate superiority over point estimators. Suppose the input-target relationship is modeled as in (2) but we write $\theta$ instead of $w$. We set $q(x) = N(0, I_3)$. For now consider the realisable case, $q(y|x) = p(y|x, \theta_0)$ where $\theta_0$ is drawn randomly according to the default initialisation in PyTorch when model (2) is instantiated. We calculate $\mathbb{E}_n G(n)$ using multiple datasets $\mathcal{D}_n$ and a large testing set, see Appendix A.5 for more details.

Since $f$ is a hierarchical model, let's write it as $f_\theta(\cdot) = h(g(\cdot; v); w)$ with the dimension of $w$ being relatively small. Let $\theta_{\text{MAP}} = (v_{\text{MAP}}, w_{\text{MAP}})$ be the MAP estimate for $\theta$ using batch gradient descent. The idea of our simple approximate Bayesian scheme is to freeze the network weights at the MAP estimate for early layers and perform approximate Bayesian inference for the final layers[2]. e.g., freeze the parameters of $g$ at $v_{\text{MAP}}$ and perform MCMC over $w$. Throughout the experiments, $g : \mathbb{R}^3 \to \mathbb{R}^3$ is a feedforward ReLU block with each hidden layer having 5 hidden units and $h : \mathbb{R}^3 \to \mathbb{R}^3$ is either $BAx$ or $B\,\text{ReLU}(Ax)$ where $A \in \mathbb{R}^{3\times r}, B \in \mathbb{R}^{r\times 3}$. We set $r = 3$. We shall consider 1 or 5 hidden layers for $g$.

To approximate the Bayes predictive distribution, we perform either the Laplace approximation or the NUTS variant of HMC (Hoffman & Gelman, 2014) in the last two layers, i.e., performing inference over $A, B$ in $h(g(\cdot; v_{\text{MAP}}); A, B)$. Note that MCMC is operating in a space of 18 di-

---

[2]This is similar in spirit to Kristiadi et al. (2020) who claim that even "being Bayesian a little bit" fixes overconfidence. They approach this via the Laplace approximation for the final layer of a ReLU network. It is also worth noting that Kristiadi et al. (2020) do not attempt to formalise what it means to "fix overconfidence"; the precise statement should be in terms of $G(n)$.

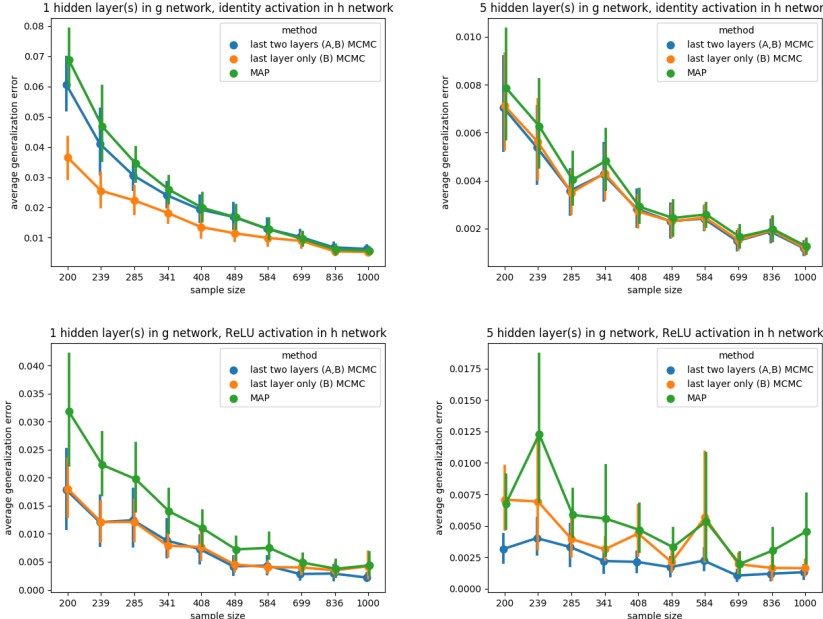

Figure 1: *Realisable and full batch gradient descent for MAP.* Average generalisation errors $\mathbb{E}_n G(n)$ are displayed for various approximations of the Bayes predictive distribution. The results of the Laplace approximations are reported in the Appendix and not displayed here because they are higher than other approximation schemes by at least an order of magnitude. Each subplot shows a different combination of hidden layers in $g$ (1 or 5) and activation function in $h$ (ReLU or identity). Note that the y-axis is not shared.

Table 1: Companion to Figure 1. The learning coefficient is the slope of the linear fit $1/n$ versus $\mathbb{E}_n G(n)$ (no intercept since realisable). The $R^2$ value gives a sense of the goodness-of-fit.

(a) 1 hidden layer(s) in $g$, identity activation in $h$

| method | learning coefficient | R squared |
|---|---|---|
| last two layers (A,B) MCMC | 9.709027 | 0.966124 |
| last layer only (B) MCMC | 6.410380 | 0.988921 |
| last two layers (A,B) Laplace | inf | NaN |
| last layer only (B) Laplace | 2154.989266 | 0.801077 |
| MAP | 10.714216 | 0.951051 |

(b) 5 hidden layer(s) in $g$, identity activation in $h$

| method | learning coefficient | R squared |
|---|---|---|
| last two layers (A,B) MCMC | 1.286290 | 0.985161 |
| last layer only (B) MCMC | 1.298504 | 0.982298 |
| last two layers (A,B) Laplace | inf | NaN |
| last layer only (B) Laplace | 2038.605589 | 0.803736 |
| MAP | 1.437473 | 0.983411 |

(c) 1 hidden layer(s) in $g$, ReLU activation in $h$

| method | learning coefficient | R squared |
|---|---|---|
| last two layers (A,B) MCMC | 3.117187 | 0.977313 |
| last layer only (B) MCMC | 3.152710 | 0.980132 |
| last two layers (A,B) Laplace | inf | NaN |
| last layer only (B) Laplace | 1120.648298 | 0.742412 |
| MAP | 5.343311 | 0.972212 |

(d) 5 hidden layer(s) in $g$, ReLU activation in $h$

| method | learning coefficient | R squared |
|---|---|---|
| last two layers (A,B) MCMC | 0.835593 | 0.957824 |
| last layer only (B) MCMC | 1.466273 | 0.920716 |
| last two layers (A,B) Laplace | inf | NaN |
| last layer only (B) Laplace | 1416.294288 | 0.808991 |
| MAP | 1.981483 | 0.889519 |

mensions in this case, which is small enough for us to expect MCMC to perform well. We also implemented the Laplace approximation and NUTS in the last layer only, i.e. performing inference over $B$ in $h_2(h_1(g(\cdot; v_{\mathrm{MAP}}); A_{\mathrm{MAP}}); B)$. Further implementation details of these approximate Bayesian schemes are found in Appendix A.5.

From the outset, we expect the Laplace approximation over $w = (A, B)$ to be invalid since the model is singular. We do however expect the last-layer-only Laplace approximation over $B$ to be sound. Next, we expect the MCMC approximation in either the last layer or last two layers to be superior to the Laplace approximations and to the MAP. We further expect the last-two-layers MCMC to have better generalisation than the last-layer-only MCMC since the former is closer to the Bayes predictive distribution. In summary, we anticipate the following performance order for

these five approximate Bayesian schemes (from worst to best): last-two-layers Laplace, last-layer-only Laplace, MAP, last-layer-only MCMC, last-two-layers MCMC.

The results displayed in Figure 1 are in line with our stated expectations above, *except* for the surprise that the last-layer-only MCMC approximation is often superior to the last-two-layers MCMC approximation. This may arise from the fact that MCMC finds the singular setting in the last-two-layers more challenging. In Figure 1, we clarify the effect of the network architecture by varying the following factors: 1) either 1 or 5 layers in $g$, and 2) ReLU or identity activation in $h$. Table 1 is a companion to Figure 1 and tabulates for each approximation scheme the slope of $1/n$ versus $\mathbb{E}_n G(n)$, also known as the learning coefficient. The $R^2$ corresponding to the linear fit is also provided. In Appendix A.5, we also show the corresponding results when 1) the data-generating mechanism and the assumed model do not satisfy the condition of realisability and/or 2) the MAP estimate is obtained via minibatch stochastic gradient descent instead of batch gradient descent.

## 6 SIMPLE FUNCTIONS AND COMPLEX SINGULARITIES

In singular models the RLCT may vary with the true distribution (in contrast to regular models) and in this section we examine this phenomenon in a simple example. As the true distribution becomes more complicated relative to the supposed model, the singularities of the analytic variety of true parameters should become simpler and hence the RLCT should increase (Watanabe, 2009, §7.6). Our experiments are inspired by (Watanabe, 2009, §7.2) where $\tanh(x)$ networks are considered and the true distribution (associated to the zero network) is held fixed while the number of hidden nodes is increased.

Consider the model $p(y|x, w)$ in (2) where $f(x, w) = c + \sum_{i=1}^{H} q_i \operatorname{ReLU}(\langle w_i, x \rangle + b_i)$ is a two-layer ReLU network with weight vector $w = (\{w_i\}_{i=1}^H, \{b_i\}_{i=1}^H, \{q_i\}_{i=1}^H, c) \in \mathbb{R}^{4H+1}$ and $w_i \in \mathbb{R}^2, b_i \in \mathbb{R}, q_i \in \mathbb{R}$ for $1 \le i \le H$. We let $W$ be some compact neighborhood of the origin.

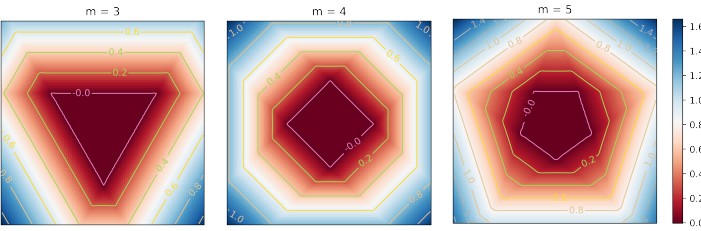

Figure 2: Increasingly complicated true distributions $q_m(x, y)$ on $[-1, 1]^2 \times \mathbb{R}$.

Table 2: RLCT estimates for ReLU and SiLU networks. We observe the RLCT increasing as $m$ increases, i.e., the true distribution becomes more "complicated" relative to the supposed model.

| m | Nonlinearity | RLCT | Std | R squared |
|---|---|---|---|---|
| 3 | ReLU | 0.526301 | 0.027181 | 0.983850 |
| 3 | SiLU | 0.522393 | 0.026342 | 0.978770 |
| 4 | ReLU | 0.539590 | 0.024774 | 0.991241 |
| 4 | SiLU | 0.539387 | 0.020769 | 0.988495 |
| 5 | ReLU | 0.555303 | 0.002344 | 0.993092 |
| 5 | SiLU | 0.555630 | 0.021184 | 0.990971 |

Given an integer $3 \le m \le H$ we define a network $s_m \in W$ and $q_m(y|x) := p(y|x, s_m)$ as follows. Let $g \in SO(2)$ stand for rotation by $2\pi/m$, set $w_1 = \sqrt{g}\,(1, 0)^T$. The components of $s_m$ are the vectors $w_i = g^{i-1} w_1$ for $1 \le i \le m$ and $w_i = 0$ for $i > m$, $b_i = -\frac{1}{3}$ and $q_i = 1$ for $1 \le i \le m$ and $b_i = q_i = 0$ for $i > m$, and finally $c = 0$. The factor of $\frac{1}{3}$ ensures the relevant parts of the decision boundaries lie within $X = [-1, 1]^2$. We let $q(x)$ be the uniform distribution on $X$ and define $q_m(x, y) = q_m(y|x)q(x)$. The functions $f(x, s_m)$ are graphed in Figure 2. It is intuitively clear that the complexity of these true distributions increases with $m$.

We let $\varphi$ be a normal distribution $N(0, 50^2)$ and estimate the RLCTs of the triples $(p, q_m, \varphi)$. We conducted the experiments with $H = 5$, $n = 1000$. For each $m \in \{3, 4, 5\}$, Table 2 shows the

estimated RLCT. Algorithm 1 in Appendix A.3 details the estimation procedure which we base on (Watanabe, 2013, Theorem 4). As predicted the RLCT increases with $m$ verifying that in this case, the simpler true distributions give rise to more complex singularities.

Note that the dimension of $W$ is $d = 21$ and so if the model were regular the RLCT would be 10.5. It can be shown that when $m = H$ the set of true parameters $W_0 \subseteq W$ is a regular submanifold of dimension $m$. If such a model were minimally singular its RLCT would be $\frac{1}{2}((4m + 1) - m) = \frac{1}{2}(3m + 1)$. In the case $m = 5$ we observe an RLCT more than an order of magnitude less than the value 8 predicted by this formula. So the function $K$ does not behave like a quadratic form near $W_0$.

Strictly speaking it is incorrect to speak of the RLCT of a ReLU network because the function $K(w)$ is not necessarily analytic (Example A.4). However we observe empirically that the predicted linear relationship between $E_w^\beta[nL_n(w)]$ and $1/\beta$ holds in our small ReLU networks (see the $R^2$ values in Table 2) and that the RLCT estimates are close to those for the two-layer SiLU network (Hendrycks & Gimpel, 2016) which is analytic (the SiLU or sigmoid weighted linear unit is $\sigma(x) = x(1 + e^{-\tau x})^{-1}$ which approaches the ReLU as $\tau \to \infty$. We use $\tau = 100.0$ in our experiments). The competitive performance of SiLU on standard benchmarks (Ramachandran et al., 2017) shows that the non-analyticity of ReLU is probably not fundamental.

## 7 FUTURE DIRECTIONS

Deep neural networks are singular models, and that's good: the presence of singularities is *necessary* for neural networks with large numbers of parameters to have low generalisation error. Singular learning theory clarifies how classical tools such as the Laplace approximation are not just inappropriate in deep learning on narrow technical grounds: the failure of this approximation and the existence of interesting phenomena like the generalisation puzzle have a common cause, namely the existence of degenerate critical points of the KL function $K(w)$. Singular learning theory is a promising foundation for a mathematical theory of deep learning. However, much remains to be done. The important open problems include:

**SGD vs the posterior.** A number of works (ŞimŞekli, 2017; Mandt et al., 2017; Smith et al., 2018) suggest that mini-batch SGD may be governed by SDEs that have the posterior distribution as its stationary distribution and this may go towards understanding why SGD works so well for DNNs.

**RLCT estimation for large networks.** Theoretical RLCTs have been cataloged for small neural networks, albeit at significant effort[3] (Aoyagi & Watanabe, 2005b;a). We believe RLCT estimation in these small networks should be standard benchmarks for any method that purports to approximate the Bayesian posterior of a neural network. No theoretical RLCTs or estimation procedure are known for modern DNNs. Although MCMC provides the gold standard it does not scale to large networks. The intractability of RLCT estimation for DNNs is not necessarily an obstacle to reaping the insights offered by singular learning theory. For instance, used in the context of model selection, the exact value of the RLCT is not as important as model selection consistency. We also demonstrated the utility of singular learning results such as (9) and (10) which can be exploited even without knowledge of the exact value of the RLCT.

**Real-world distributions are unrealisable.** The existence of power laws in neural language model training (Hestness et al., 2017; Kaplan et al., 2020) is one of the most remarkable experimental results in deep learning. These power laws may be a sign of interesting new phenomena in singular learning theory when the true distribution is unrealisable.

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

## A APPENDIX

### A.1 NEURAL NETWORKS ARE STRICTLY SINGULAR

Many-layered neural networks are strictly singular (Watanabe, 2009, §7.2). The degeneracy of the Hessian in deep learning has certainly been acknowledged in e.g., Sagun et al. (2016) which recognises the eigenspectrum is concentrated around zero and in Pennington & Worah (2018) which deliberately studies the Fisher information matrix of a *single*-hidden-layer, rather than multilayer, neural network.

We first explain how to think about a neural network in the context of singular learning theory. A feedforward network of depth $c$ parametrises a function $f : \mathbb{R}^N \longrightarrow \mathbb{R}^M$ of the form

$$f = A_c \circ \sigma_{c-1} \circ A_{c-1} \cdots \sigma_1 \circ A_1$$

where the $A_l : \mathbb{R}^{d_{l-1}} \longrightarrow \mathbb{R}^{d_l}$ are affine functions and $\sigma_l : \mathbb{R}^{d_l} \longrightarrow \mathbb{R}^{d_l}$ is coordinate-wise some fixed nonlinearity $\sigma : \mathbb{R} \longrightarrow \mathbb{R}$. Let $W$ be a compact subspace of $\mathbb{R}^d$ containing the origin, where $\mathbb{R}^d$ is the space of sequences of affine functions $(A_l)_{l=1}^c$ with coordinates denoted $w_1, \ldots, w_d$ so that $f$ may be viewed as a function $f : \mathbb{R}^N \times W \longrightarrow \mathbb{R}^M$. We define $p(y|x, w)$ as in (2). We assume the true distribution is realisable, $q(y|x) = p(y|x, w_0)$ and that a distribution $q(x)$ on $\mathbb{R}^N$ is fixed with respect to which $p(x, y) = p(y|x)q(x)$ and $q(x, y) = q(y|x)q(x)$. Given some prior $\varphi(w)$ on $W$ we may apply singular learning theory to the triplet $(p, q, \varphi)$.

By straightforward calculations we obtain

$$K(w) = \tfrac{1}{2} \int \|f(x, w) - f(x, w_0)\|^2 q(x) dx \tag{11}$$

$$\frac{\partial^2}{\partial w_i \partial w_j} K(w) = \int \left\langle \frac{\partial}{\partial w_i} f(x, w), \frac{\partial}{\partial w_j} f(x, w) \right\rangle q(x) dx$$

$$+ \int \left\langle f(x, w) - f(x, w_0), \frac{\partial^2}{\partial w_i \partial w_j} f(x, w) \right\rangle q(x) dx \tag{12}$$

$$I(w)_{ij} = \frac{1}{2^{(M-3)/2} \pi^{(M-2)/2}} \int \left\langle \frac{\partial}{\partial w_i} f(x, w), \frac{\partial}{\partial w_j} f(x, w) \right\rangle q(x) dx \tag{13}$$

where $\langle -, - \rangle$ is the dot product. We assume $q(x)$ is such that these integrals exist.

It will be convenient below to introduce another set of coordinates for $W$. Let $w_{jk}^l$ denote the weight from the $k$th neuron in the $(l-1)$th layer to the $j$th neuron in the $l$th layer and let $b_j^l$ denote the bias of the $j$th neuron in the $l$th layer. Here $1 \leq l \leq c$ and the input is layer zero. Let $u_j^l$ and $a_j^l$ denote the value of the $j$th neuron in the $l$th layer before and after activation, respectively. Let $u^l$ and $a^l$ denote the vectors with values $u_j^l$ and $a_j^l$, respectively. Let $d_l$ denote the number of neurons in the $l$th layer. Then

$$u_j^l = \sum_{k=1}^{d_{l-1}} w_{jk}^l a_k^{l-1} + b_j^l, \qquad\qquad 1 \leq l \leq c, 1 \leq j \leq d_l$$

$$a_j^l = \sigma(u_j^l) \qquad\qquad 1 \leq l < c, 1 \leq j \leq d_l$$

with the convention that $a^0 = x$ is the input and $u^c = y$ is the output.

In the case where $\sigma = \mathrm{ReLU}$ the partial derivatives $\frac{\partial}{\partial w_j} f$ do not exist on all of $\mathbb{R}^N$. However given $w \in W$ we let $\mathcal{D}(w)$ denote the complement in $\mathbb{R}^N$ of the union over all hidden nodes of the associated decision boundary, that is

$$\mathbb{R}^N \setminus \mathcal{D}(w) = \bigcup_{1 \leq l < c} \bigcup_{1 \leq j \leq d_l} \{x \in \mathbb{R}^N : u_j^l(x) = 0\}.$$

The partial derivative $\frac{\partial}{\partial w_j} f$ exists on the open subset $\{(x, w) : x \in \mathcal{D}(w)\}$ of $\mathbb{R}^N \times W$.

**Lemma A.1.** *Suppose $\sigma = \mathrm{ReLU}$ and there are $c > 1$ layers. For any hidden neuron $1 \leq j \leq d_l$ in layer $l$ with $1 \leq l < c$ there is a differential equation*

$$\left\{ \sum_{k=1}^{d_{l-1}} w_{jk}^l \frac{\partial}{\partial w_{jk}^l} + b_j^l \frac{\partial}{\partial b_j^l} - \sum_{i=1}^{d_{l+1}} w_{ij}^{l+1} \frac{\partial}{\partial w_{ij}^{l+1}} \right\} f = 0$$

*which holds on $\mathcal{D}(w)$ for any fixed $w \in W$.*

*Proof.* Without loss of generality assume $M = 1$, to simplify the notation. Let $e_i \in \mathbb{R}^{d_{l+1}}$ denote a unit vector and let $H(x) = \frac{d}{dx} \mathrm{ReLU}(x)$. Writing $\frac{\partial f}{\partial u^{l+1}}$ for a gradient vector

$$\frac{\partial f}{\partial w_{ij}^{l+1}} = \left\langle \frac{\partial f}{\partial u^{l+1}}, \frac{\partial u^{l+1}}{\partial w_{ij}^{l+1}} \right\rangle = \left\langle \frac{\partial f}{\partial u^{l+1}}, a_j^l e_i \right\rangle = \frac{\partial f}{\partial u_i^{l+1}} u_j^l H(u_j^l)$$

$$\frac{\partial f}{\partial w_{jk}^l} = \left\langle \frac{\partial f}{\partial u^{l+1}}, \frac{\partial u^{l+1}}{\partial w_{jk}^l} \right\rangle = \left\langle \frac{\partial f}{\partial u^{l+1}}, \sum_{i=1}^{d_{l+1}} w_{ij}^{l+1} a_k^{l-1} H(u_j^l) e_i \right\rangle = \sum_{i=1}^{d_{l+1}} \frac{\partial f}{\partial u_i^{l+1}} w_{ij}^{l+1} a_k^{l-1} H(u_j^l)$$

$$\frac{\partial f}{\partial b_j^l} = \left\langle \frac{\partial f}{\partial u^{l+1}}, \frac{\partial u^{l+1}}{\partial b_j^l} \right\rangle = \left\langle \frac{\partial f}{\partial u^{l+1}}, \sum_{i=1}^{d_{l+1}} w_{ij}^{l+1} H(u_j^l) e_i \right\rangle = \sum_{i=1}^{d_{l+1}} \frac{\partial f}{\partial u_i^{l+1}} w_{ij}^{l+1} H(u_j^l).$$

The claim immediately follows. $\qquad\square$

**Lemma A.2.** *Suppose $\sigma = \mathrm{ReLU}, c > 1$ and that $w \in W$ has at least one weight or bias at a hidden node nonzero. Then the matrix $I(w)$ is degenerate and if $w \in W_0$ then the Hessian of $K$ at $w$ is also degenerate.*

*Proof.* Let $w \in W$ be given, and choose a hidden node where at least one of the incident weights (or bias) is nonzero. Then Lemma A.1 gives a nontrivial linear dependence relation $\sum_i \lambda_i \frac{\partial}{\partial w_i} f = 0$ as functions on $\mathcal{D}(w)$. The rows of $I(w)$ satisfy the same linear dependence relation. At a true parameter the second summand in (12) vanishes so by the same argument the Hessian is degenerate. $\qquad\square$

**Remark A.3.** Lemma A.2 implies that every true parameter for a nontrivial ReLU network is a degenerate critical point of $K$. Hence in the study of nontrivial ReLU networks it is never appropriate to divide by the determinant of the Hessian of $K$ at a true parameter, and in particular Laplace or saddle-point approximations at a true parameter are invalid.

The well-known positive scale invariance of ReLU networks (Phuong & Lampert, 2020) is responsible for the linear dependence of Lemma A.1, in the precise sense that the given differential operator is the infinitesimal generator (Boothby, 1986, §IV.3) of the scaling symmetry. However, this is only one source of degeneracy or singularity in ReLU networks. The degeneracy, as measured by the RLCT, is much lower than one would expect on the basis of this symmetry alone (see Section 6).

**Example A.4.** In general the KL function $K(w)$ for ReLU networks is not analytic. For the minimal counterexample, let $q(x)$ be uniform on $[-N, N]$ and zero outside and consider

$$K(b) = \int q(x)(\mathrm{ReLU}(x - b) - \mathrm{ReLU}(x))^2 dx.$$

It is easy to check that up to a scalar factor

$$K(b) = \begin{cases} -\frac{2}{3}b^3 + b^2 N & 0 \le b \le N \\ -\frac{1}{3}b^3 + b^2 N & -N \le b \le 0 \end{cases}$$

so that $K$ is $C^2$ but not $C^3$ let alone analytic.

## A.2 REDUCED RANK REGRESSION

For reduced rank regression, the model is

$$p(y|x, w) = \frac{1}{(2\pi\sigma^2)^{N/2}} \exp\left(-\frac{1}{2\sigma^2}|y - BAx|^2\right),$$

where $x \in \mathbb{R}^M, y \in \mathbb{R}^N$, $A$ an $M \times H$ matrix and $B$ an $H \times N$ matrix; the parameter $w$ denotes the entries of $A$ and $B$, i.e. $w = (A, B)$, and $\sigma > 0$ is a parameter which for the moment is irrelevant.

If the true distribution is realisable then there is $w_0 = (A_0, B_0)$ such that $q(y|x) = p(y|x, w_0)$. Without loss of generality assume $q(x)$ is the uniform density. In this case the KL divergence from $p(y|x, w)$ to $q(y|x)$ is

$$K(w) = \int q(y|x) \log \frac{q(y|x)}{p(y|x, w)} dx dy = \|BA - B_0 A_0\|^2 (1 + E(w))$$

where the error $E$ is smooth and $E(w) = O(\|BA - B_0 A_0\|^2)$ in any region where $\|BA - B_0 A_0\| < C$, so $K(w)$ is equivalent to $\|BA - B_0 A_0\|^2$. We write $K(w) = \|BA - B_0 A_0\|^2$ for simplicity below.

Now assume that $B_0 A_0$ is symmetric and that $B_0$ is square, i.e. $N = H$. Then the zero locus of $K(w)$ is explicitly given as follows

$$W_0 = \{(A, B) : \det B \ne 0 \text{ and } A = B^{-1} B_0 A_0\}.$$

It follows that $W_0$ is globally a graph over $GL(H; \mathbb{R})$. Indeed, the set $(B^{-1} B_0 A_0, B)$ with $B \in GL(H; \mathbb{R})$ is exactly $W_0$. Thus $W_0$ is a smooth $H^2$-dimensional submanifold of $\mathbb{R}^{H^2} \times \mathbb{R}^{H \times M}$. To prove that $W_0$ is minimally singular in the sense of Section 4 it suffices to show that $\mathrm{rank}(D^2_{A,B} K) \ge HM$ where $D^2_{A,B} K$ denotes the Hessian, but as it is no more difficult to do so, we find explicit local coordinates $(u, v)$ near an arbitrary point $(\overline{A}, \overline{B}) \in W_0$ for which $\{v = 0\} = W_0$ and $K(u, v) = a(u, v)|u|^2$ in this neighborhood, where $a$ is a $C^\infty$ function with $a \ge c > 0$ for some $c$. Write

$$A(v) = (\overline{B} + v)^{-1} B_0 A_0.$$

Then $u, v \mapsto (A(v) + u, \overline{B} + v)$ gives local coordinates on $\mathbb{R}^{H^2} \times \mathbb{R}^{H \times M}$ near $(\overline{A}, \overline{B})$, and

$$K(u, v) = |(\overline{B} + v)((\overline{B} + v)^{-1} B_0 A_0 + u) - B_0 A_0|$$
$$= |B_0 A_0 + (\overline{B} + v)u - B_0 A_0|^2$$
$$= |(\overline{B} + v)u|^2,$$

so for $v$ sufficiently small (and hence $\overline{B} + v$ invertible) we can take $a(u, v) = |(\overline{B} + v)u|^2/|u|^2$.

## A.3 RLCT ESTIMATION

In this section we detail the estimation procedure for the RLCT used in Section 6. Let $L_n(w)$ be the negative log likelihood as in (3). Define the data likelihood at inverse temperature $\beta > 0$ to be

---

**Algorithm 1** RLCT via Theorem 4 in Watanabe (2013)

---

**Input:** range of $\beta$'s, set of training sets $\mathcal{T}$ each of size $n$, approximate samples $\{w_1, \ldots, w_R\}$ from $p^\beta(w|\mathcal{D}_n)$ for each training set $\mathcal{D}_n$ and each $\beta$
**for** training set $\mathcal{D}_n \in \mathcal{T}$ **do**
   **for** $\beta$ in range of $\beta$'s **do**
      Approximate $\mathbb{E}_w^\beta[nL_n(w)]$ with $\frac{1}{R}\sum_{i=1}^R nL_n(w_r)$ where $w_1, \ldots, w_R$ are approximate samples from $p^\beta(w|\mathcal{D}_n)$
   **end for**
   Perform generalised least squares to fit $\lambda$ in (18), call result $\hat{\lambda}(\mathcal{D}_n)$
**end for**
**Output:** $\frac{1}{|\mathcal{T}|}\sum_{\mathcal{D}_n \in \mathcal{T}} \hat{\lambda}(\mathcal{D}_n)$

---

$p^\beta(\mathcal{D}_n|w) = \Pi_{i=1}^n p(y_i|x_i, w)^\beta$ which can also be written

$$p^\beta(\mathcal{D}_n|w) = \exp(-\beta n L_n(w)). \tag{14}$$

The posterior distribution, at inverse temperature $\beta$, is defined as

$$p^\beta(w|\mathcal{D}_n) = \frac{\Pi_{i=1}^n p(y_i|x_i, w)^\beta \varphi(w)}{\int_W \Pi_{i=1}^n p(y_i|x_i, w)^\beta \varphi(w)} = \frac{p^\beta(\mathcal{D}_n|w)\varphi(w)}{p^\beta(\mathcal{D}_n)} \tag{15}$$

where $\varphi$ is the prior distribution on the network weights $w$ and

$$p^\beta(\mathcal{D}_n) = \int_W p^\beta(\mathcal{D}_n|w)\varphi(w)\,dw \tag{16}$$

is the marginal likelihood of the data at inverse temperature $\beta$. Finally, denote the expectation of a random variable $R(w)$ with respect to the tempered posterior $p^\beta(w|\mathcal{D}_n)$ as

$$\mathbb{E}_w^\beta[R(w)] = \int_W R(w)p^\beta(w|\mathcal{D}_n)\,dw \tag{17}$$

In the main text, we drop the superscript in the quantities (14), (15), (16), (17) when $\beta = 1$, e.g., $p(\mathcal{D}_n)$ rather than $p^1(\mathcal{D}_n)$.

Assuming the conditions of Theorem 4 in Watanabe (2013) hold, we have

$$\mathbb{E}_w^\beta[nL_n(w)] = nL_n(w_0) + \frac{\lambda}{\beta} + U_n\sqrt{\frac{\lambda}{2\beta}} + O_p(1) \tag{18}$$

where $\beta_0$ is a positive constant and $U_n$ is a sequence of random variables satisfying $\mathbb{E}_n U_n = 0$. In Algorithm 1, we describe an estimation procedure for the RLCT based on the asymptotic result in (18).

For the estimates in Table 2 the *a posteriori* distribution was approximated using the NUTS variant of Hamiltonian Monte Carlo (Hoffman & Gelman, 2014) where the first 1000 steps were omitted and $20,000$ samples were collected. Each $\hat{\lambda}(\mathcal{D}_n)$ estimate in Algorithm 1 was performed by linear regression on the pairs $\{(1/\beta_i, \mathbb{E}_w^{\beta_i}[nL_n(w)])\}_{i=1}^5$ where the five inverse temperatures $\beta_i$ are centered on the inverse temperature $1/\log(20000)$.

## A.4 CONNECTION BETWEEN RLCT AND GENERALISATION

For completeness, we sketch the derivation of (9) which gives the asymptotic expansion of the average generalisation error $\mathbb{E}_n G(n)$ of the Bayes prediction distribution in singular models. The exposition is an amalgamation of various works published by Sumio Watanabe, but is mostly based on the textbook (Watanabe, 2009).

To understand the connection between the RLCT and $G(n)$, we first define the so-called **Bayes free energy** as

$$F(n) = -\log p(\mathcal{D}_n)$$

whose expectation admits the following asymptotic expansion (Watanabe, 2009):

$$\mathbb{E}_n F(n) = \mathbb{E}_n n S_n + \lambda \log n + o(\log n)$$

where $S_n = -\frac{1}{n}\sum_{i=1}^n \log q(y_i|x_i)$ is the entropy. The expected Bayesian generalisation error is related to the Bayes free energy as follows

$$\mathbb{E}_n G(n) = \mathbb{E}F(n+1) - \mathbb{E}F(n)$$

Then for the average generalisation error, we have

$$\mathbb{E}_n G(n) = \lambda/n + o(1/n). \tag{19}$$

Since models with more complex singularities have smaller RLCTs, this would suggest that the more singular a model is, the better its generalisation (assuming one uses the Bayesian predictive distribution for prediction). In this connection it is interesting to note that simpler (relative to the model) true distributions lead to more singular models (Section 6).

### A.5 DETAILS FOR GENERALISATION ERROR EXPERIMENTS

**Simulated data** The distribution of $x \in \mathbb{R}^3$ is set to $q(x) = N(0, I_3)$. In the realisable case, $y \in \mathbb{R}^3$ is drawn according to $q(y|x) = p(y|x, \theta_0)$. In the nonrealisable setting, we set $q(y|x) \propto \exp\{-||y - h_{w_0}(x)||^2/2\}$, where $w_0 = (A_0, B_0)$ is drawn according to the PyTorch model initialisation of $h$.

**MAP training** The MAP estimator is found via gradient descent using the mean-squared-error loss with either the full data set or minibatch set to 32. Training was set to 5000 epochs. No form of early stopping was employed.

**Calculating the generalisation error** Using a held-out-test set $T_{n'} = \{(x_i', y_i')\}_{i=1}^{n'}$, we calculate the average generalisation error as

$$\frac{1}{n'}\sum_{i=1}^{n'} \log q(y_i'|x_i') - \mathbb{E}_n \frac{1}{n'}\sum_{i=1}^{n'} \log \hat{q}_n(y_i'|x_i') \tag{20}$$

Assume the held-out test set is large enough so that the difference between $\mathbb{E}_n G(n)$ and (20) is negligible. We will refer to them interchangeably as the average generalisation error. In our experiments we use $n' = 10,000$ and 30 draws of the dataset $\mathcal{D}_n$ to estimate $\mathbb{E}_n$.

**Last layer(s) inference** Without loss of generality, we discuss performing inference in the $w$ parameters of $h$ while freezing the parameters of $g$ at the MAP estimate. The steps easily extend to performing inference over the final layer only of $f = h \circ g$. Let $\tilde{x}_i = g_{v_{\text{MAP}}}(x_i)$. Define a new transformed dataset $\tilde{\mathcal{D}}_n = \{(\tilde{x}_i, y_i)\}_{i=1}^n$. We take the prior on $w$ to be standard Gaussian. Define the posterior over $w$ given $\tilde{\mathcal{D}}_n$ as:

$$p(w|\tilde{\mathcal{D}}_n) \propto p(\tilde{\mathcal{D}}_n|w)\varphi(w) = \Pi_{i=1}^n \exp\{-||y_i - h_w(\tilde{x}_i)||^2/2\}\varphi(w) \tag{21}$$

Define the following approximation to the Bayesian predictive distribution

$$\tilde{p}(y|x, \mathcal{D}_n) = \int p(y|x, (v_{\text{MAP}}, w))p(w|\tilde{\mathcal{D}}_n)\,dw.$$

Let $w_1, \ldots, w_R$ be some approximate samples from $p(w|\tilde{\mathcal{D}}_n)$. Then we approximate $\tilde{p}(y|x, \mathcal{D}_n)$ with

$$\frac{1}{R}\sum_{r=1}^R p(y|x, (v_{\text{MAP}}, w_r))$$

where $R$ is a large number, set to 1000 in our experiments. We consider the Laplace approximation and the NUTS variant of HMC for drawing samples from $p(w|\tilde{\mathcal{D}}_n)$:

- **Laplace in the last layer(s)** Recall $\theta_{\text{MAP}} = (v_{\text{MAP}}, w_{\text{MAP}})$ is the MAP estimate for $f_\theta$ trained with the data $\mathcal{D}_n$. With the Laplace approximation, we draw $w_1, \ldots w_R$ from the Gaussian

  $$N(w_{\text{MAP}}, \Sigma)$$

  where $\Sigma = (-\nabla^2 \log p(w|\tilde{\mathcal{D}}_n)|_{w_{\text{MAP}}})^{-1}$ is the inverse Hessian[4] of the negative log posterior evaluated at the MAP estimate of the mode.

- **MCMC in the last layer(s)** We used the NUTS variant of HMC to draw samples from (21) with the first 1000 samples discarded.. Our implementation used the `pyro` package in `PyTorch`.

---

[4]Following Kristiadi et al. (2020), the code for the exact Hessian calculation is borrowed from `https://github.com/f-dangel/hbp`

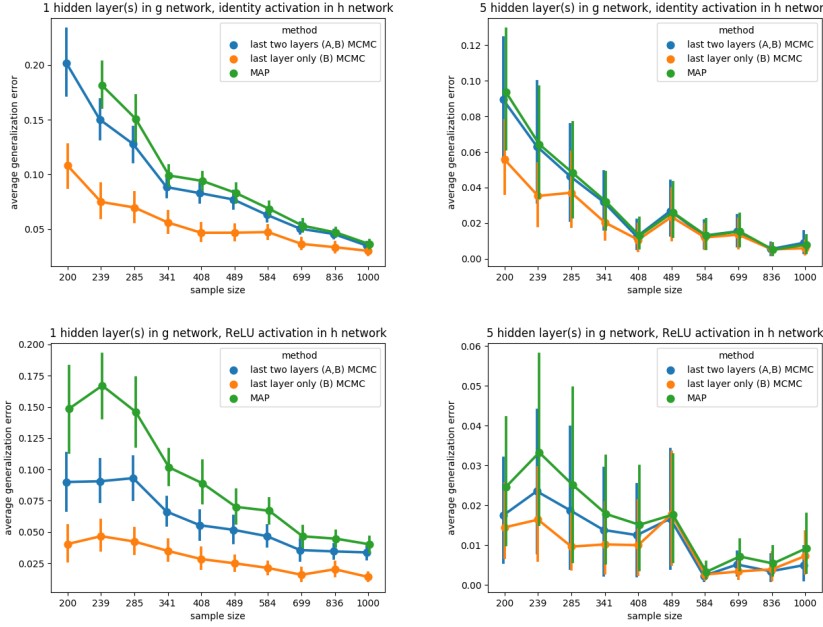

Figure 3: *Realisable and minibatch gradient descent for MAP training.*

Table 3: Companion to Figure 3.

(a) 1 hidden layer(s) in $g$, identity activation in $h$

| method | learning coefficient | R squared |
|---|---|---|
| last two layers (A,B) MCMC | 36.721594 | 0.992839 |
| last layer only (B) MCMC | 20.676920 | 0.983695 |
| last two layers (A,B) Laplace | inf | NaN |
| last layer only (B) Laplace | 1768.655088 | 0.838035 |
| MAP | inf | NaN |

(b) 5 hidden layer(s) in $g$, identity activation in $h$

| method | learning coefficient | R squared |
|---|---|---|
| last two layers (A,B) MCMC | 13.729278 | 0.924049 |
| last layer only (B) MCMC | 9.170642 | 0.945613 |
| last two layers (A,B) Laplace | inf | NaN |
| last layer only (B) Laplace | 1943.793236 | 0.794679 |
| MAP | 14.123308 | 0.917502 |

(c) 1 hidden layer(s) in $g$, ReLU activation in $h$

| method | learning coefficient | R squared |
|---|---|---|
| last two layers (A,B) MCMC | 22.175448 | 0.975450 |
| last layer only (B) MCMC | 10.675455 | 0.968584 |
| last two layers (A,B) Laplace | inf | NaN |
| last layer only (B) Laplace | inf | NaN |
| MAP | 35.647464 | 0.983284 |

(d) 5 hidden layer(s) in $g$, ReLU activation in $h$

| method | learning coefficient | R squared |
|---|---|---|
| last two layers (A,B) MCMC | 4.652483 | 0.922693 |
| last layer only (B) MCMC | 3.533366 | 0.862125 |
| last two layers (A,B) Laplace | inf | NaN |
| last layer only (B) Laplace | 1004.852367 | 0.901899 |
| MAP | 6.256696 | 0.940437 |

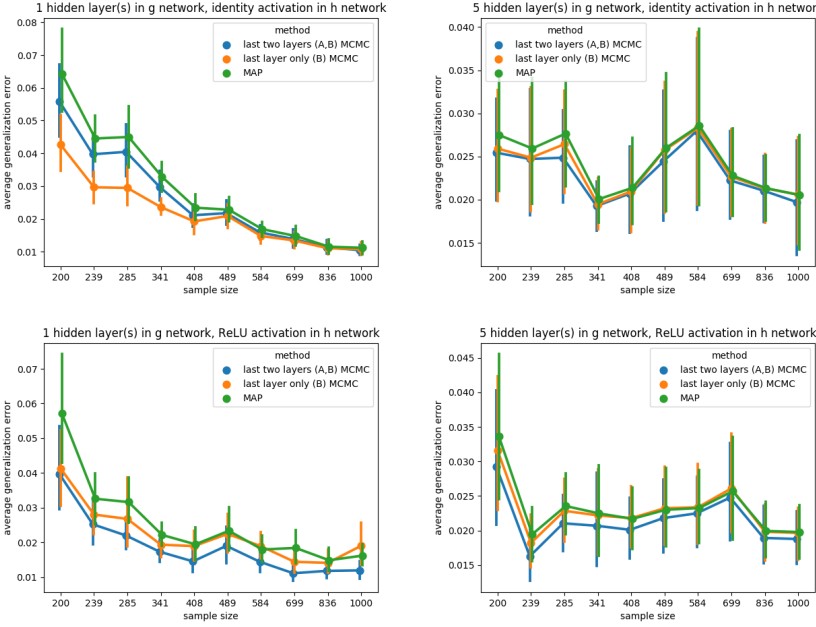

Figure 4: *Nonrealisable and full batch gradient descent for MAP training.*

Table 4: Companion to Figure 4. The learning coefficient is the slope of the linear fit $1/n$ versus $\mathbb{E}_n G(n)$ (with intercept since nonrealisable).

(a) 1 hidden layer(s) in $g$, identity activation in $h$

| method | learning coefficient | R squared |
|---|---|---|
| last two layers (A,B) MCMC | 11.086023 | 0.969991 |
| last layer only (B) MCMC | 7.377871 | 0.957824 |
| last two layers (A,B) Laplace | NaN | NaN |
| last layer only (B) Laplace | 30.692954 | 0.029238 |
| MAP | 12.947959 | 0.970173 |

(b) 5 hidden layer(s) in $g$, identity activation in $h$

| method | learning coefficient | R squared |
|---|---|---|
| last two layers (A,B) MCMC | 0.808601 | 0.144260 |
| last layer only (B) MCMC | 0.799114 | 0.127686 |
| last two layers (A,B) Laplace | NaN | NaN |
| last layer only (B) Laplace | -33.817429 | 0.009074 |
| MAP | 1.204743 | 0.242671 |

(c) 1 hidden layer(s) in $g$, ReLU activation in $h$

| method | learning coefficient | R squared |
|---|---|---|
| last two layers (A,B) MCMC | 5.987187 | 0.848490 |
| last layer only (B) MCMC | 5.384686 | 0.801313 |
| last two layers (A,B) Laplace | NaN | NaN |
| last layer only (B) Laplace | 38.629167 | 0.059012 |
| MAP | 8.560722 | 0.816794 |

(d) 5 hidden layer(s) in $g$, ReLU activation in $h$

| method | learning coefficient | R squared |
|---|---|---|
| last two layers (A,B) MCMC | 0.794055 | 0.088305 |
| last layer only (B) MCMC | 1.141580 | 0.162585 |
| last two layers (A,B) Laplace | NaN | NaN |
| last layer only (B) Laplace | -5.682602 | 0.000365 |
| MAP | 1.648073 | 0.284088 |

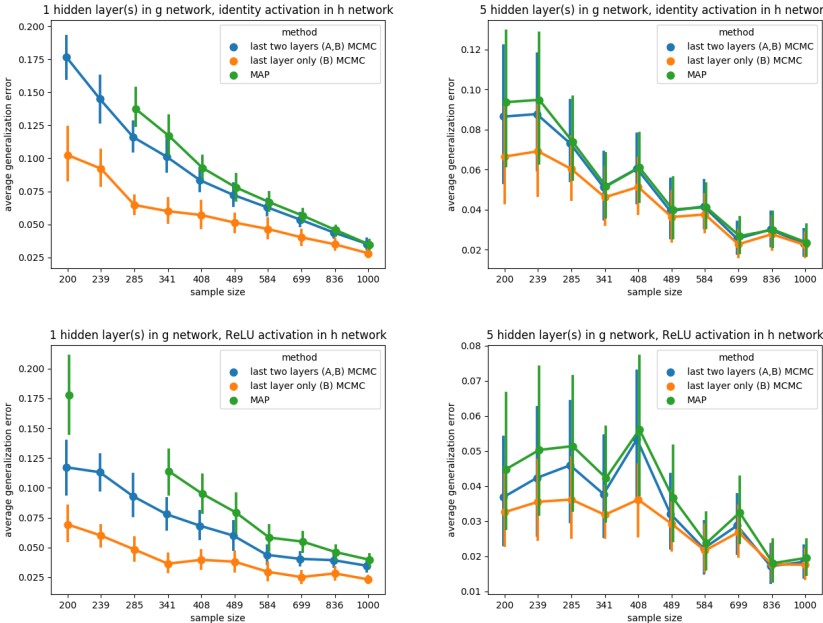

Figure 5: *Nonrealisable and minibatch gradient descent for MAP training.* Missing points on the MAP learning curve are due to estimated probabilities too close to 0.

Table 5: Companion to Figure 5. The learning coefficient is the slope of the linear fit $1/n$ versus $\mathbb{E}_n G(n)$ (with intercept since nonrealisable).

(a) 1 hidden layer(s) in $g$, identity activation in $h$

| method | learning coefficient | R squared |
|---|---|---|
| last two layers (A,B) MCMC | 11.086023 | 0.969991 |
| last layer only (B) MCMC | 7.377871 | 0.957824 |
| last two layers (A,B) Laplace | NaN | NaN |
| last layer only (B) Laplace | 30.692954 | 0.029238 |
| MAP | 12.947959 | 0.970173 |

(b) 5 hidden layer(s) in $g$, identity activation in $h$

| method | learning coefficient | R squared |
|---|---|---|
| last two layers (A,B) MCMC | 0.808601 | 0.144260 |
| last layer only (B) MCMC | 0.799114 | 0.127686 |
| last two layers (A,B) Laplace | NaN | NaN |
| last layer only (B) Laplace | -33.817429 | 0.009074 |
| MAP | 1.204743 | 0.242671 |

(c) 1 hidden layer(s) in $g$, ReLU activation in $h$

| method | learning coefficient | R squared |
|---|---|---|
| last two layers (A,B) MCMC | 5.987187 | 0.848490 |
| last layer only (B) MCMC | 5.384686 | 0.801313 |
| last two layers (A,B) Laplace | NaN | NaN |
| last layer only (B) Laplace | 38.629167 | 0.059012 |
| MAP | 8.560722 | 0.816794 |

(d) 5 hidden layer(s) in $g$, ReLU activation in $h$

| method | learning coefficient | R squared |
|---|---|---|
| last two layers (A,B) MCMC | 0.794055 | 0.088305 |
| last layer only (B) MCMC | 1.141580 | 0.162585 |
| last two layers (A,B) Laplace | NaN | NaN |
| last layer only (B) Laplace | -5.682602 | 0.000365 |
| MAP | 1.648073 | 0.284088 |

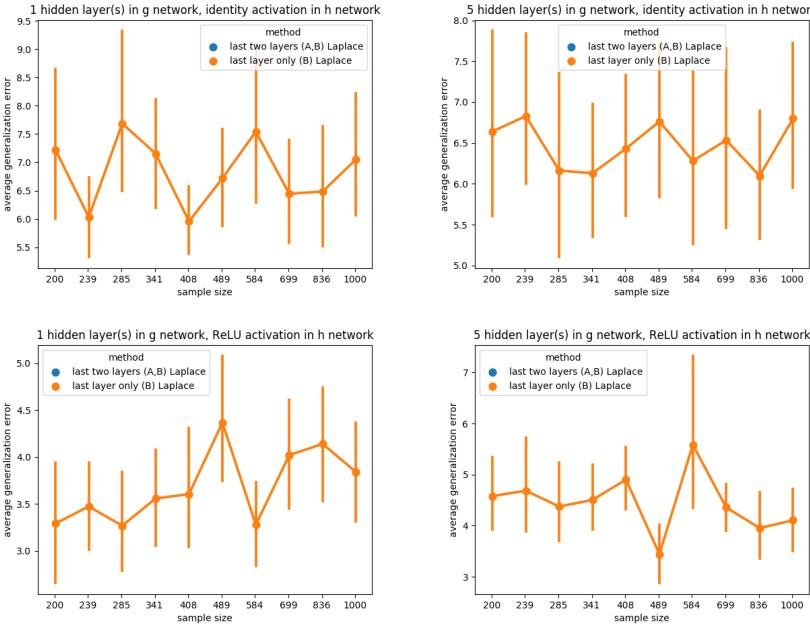

Figure 6: *Realisable and full batch gradient descent for MAP.* average generalisation errors of Laplace approximations of the predictive distribution. The last-two-layers Laplace approximation results in numerical instabilities due to degenerate Hessian. Any missing points are due to estimated probabilities too close to 0.

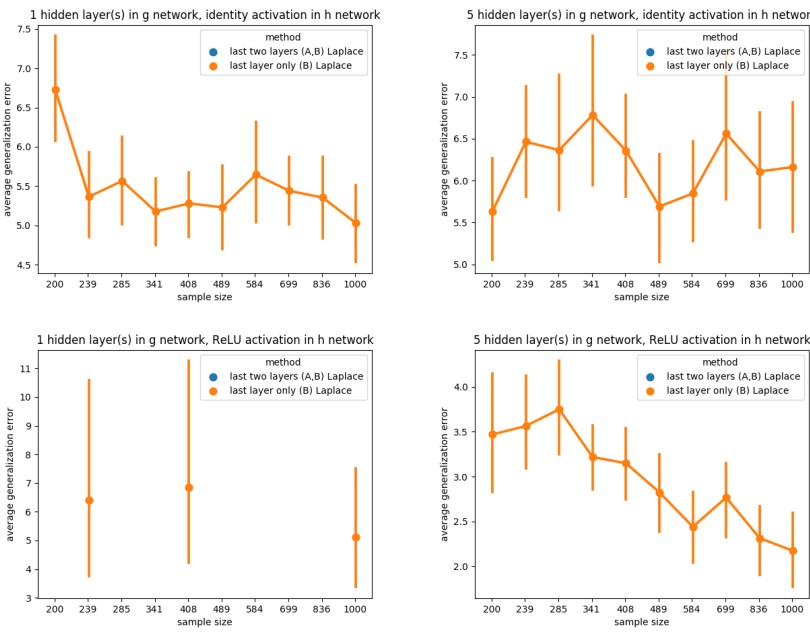

Figure 7: *Realisable and minibatch gradient descent for MAP training.* Details are same as for Figure 6

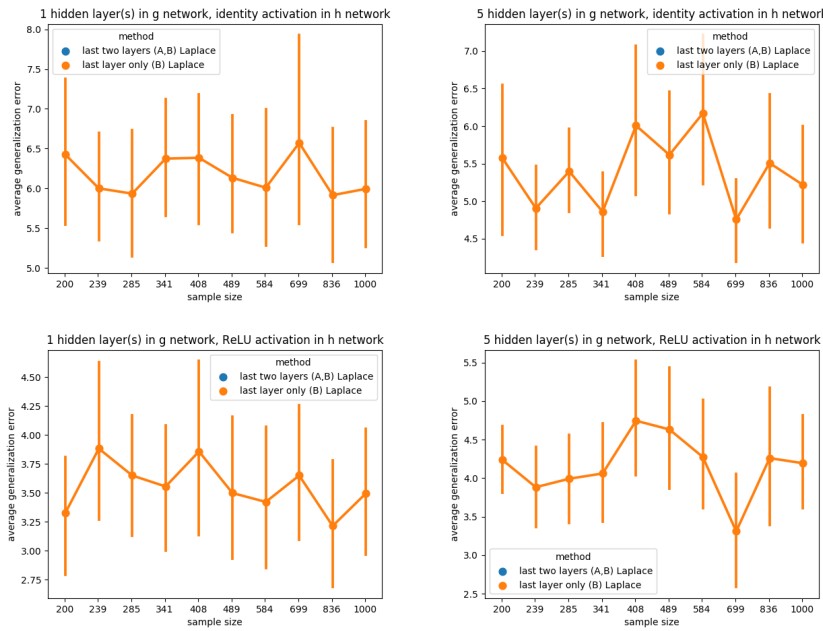

Figure 8: *Nonrealisable and full batch gradient descent for MAP training.* Details are same as for Figure 6

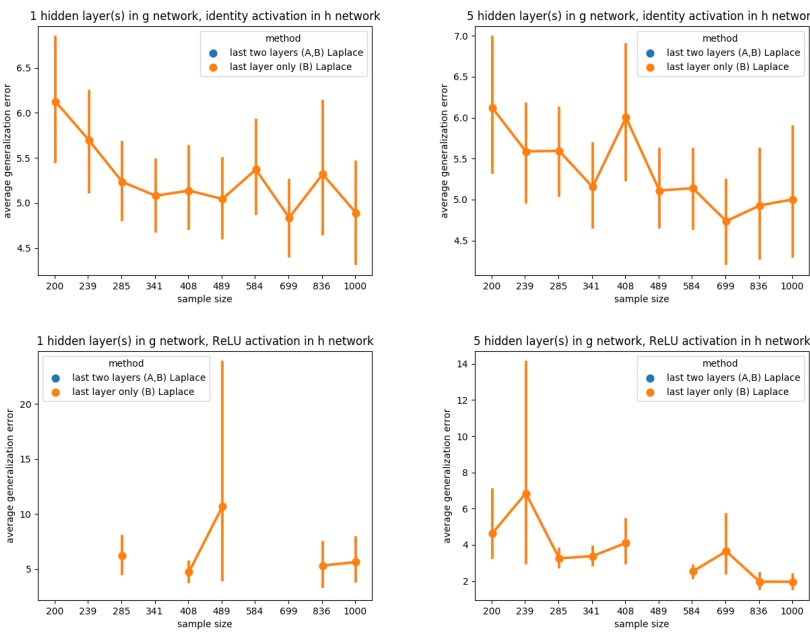

Figure 9: *Nonrealisable and minibatch gradient descent for MAP training.* Details are same as for Figure 6

