# OpenReview forum: "Deep Learning is Singular, and That's Good"
_ICLR.cc/2021/Conference — Reject_

### Official Review · AnonReviewer4 · 2020-10-28
**A review of singular learning theory**

**Rating:** 4
**Confidence:** 1

**Review:**

This paper is more like a review of singular learning theory and its implication on deep learning. The authors point out that deep neural networks are singular models and ways to characterize generalization error for regular models cannot produce satisfactory results in this setting. Then the authors introduce the singular learning theory, which has been developed for decades. Then, a series of topics for deep learning, such as flatness and generalization, are studied within the framework singular learning theory, with a combination of theoretical analysis and numerical experiments. The paper is clearly written and well organized.

pro:
The authors point out that the study of deep learning should be put into the framework of singular learning theory. They verified this point from different aspects, where it is shown that results drawn from singular learning theory is better than those drawn from regular learning theory.

con:
It seems most results in the paper are illustration or clarification of existing results.

---

> ### Author Response · Authors · 2020-11-17
> **Response to AnonReviewer4**
>
> Yes, the paper is one-part review of singular learning theory and other-part drawing the implications of singular learning theory for DNN. Thank you for noting the paper is clearly written and well-organized.
>
> Regarding the stated con, we do not agree with a scientific standard that puts novelty above all other values. If a problem is widely acknowledged to be important and has remained open despite much attention, then its solution is significant, even if that solution makes use of existing methods. For instance, the link between "flatness" and generalisation error, and how to measure the former, has occupied many published works. While short, the material in Section 4 definitively refutes this connection using singular learning theory, by explaining how the codimension of the set of true parameters (or more precisely, the RLCT) is much more important than curvature. If you wish to disagree with this conclusion or find our argument unconvincing, then we welcome that discussion, but given the amount of ink that has been spilled in the literature on this topic we cannot agree that resolving this confusion is insignificant.

---

### Official Review · AnonReviewer2 · 2020-10-28
**The paper studies the connection of Deep Learning to Singular Learning Theory but falls short in convincing why is this the right perspective.**

**Rating:** 4
**Confidence:** 3

**Review:**

The paper studies the connection of Deep Learning to Singular Learning Theory making the claim that the later could be a good foundation of the theory of Neural Networks as they are singular models.

+++++++

While I enjoyed the primer to singular learning theory, I found the paper's contribution marginal given the related work. The claim 'deep learning is singular' has already been claimed by Watanabe 07' in a much more general statement.  Moreover, each of the claims the authors made is unconvincing both from a theoretical and experimental perspectives.
For instance, the claim that RLCT governs the effective number of parameters. While the math looks sound to me, it has both has many strong assumptions on the model and lack of novelty given Watanbe 09'. To show that the calculation is meaningful, I would suggest showing the relationship to a real neural network and see how close the estimate to the real number of parameters (E.g,. in the random features (Rahimi & Recht 2008) model, the number of features is a good way to measure the effective number of parameters. Another way to estimate the real number of parameters is the location of the double descent peak (Belkin 18').

Regarding the empirical claims, the experiments are not convincing. In order to make a claim about deep learning, (say the bayes predictive error is superior to MAP or MLE) there should be either an extensive experimental demonstration of the claim (not two experiments) with proper ablation of when the claim fails (not to mention that the std of the experimental results puts all the claims to question). Same comments apply to the last section.

To conclude, my main comment is that while Deep Learning is singular (which is not a deep claim) to make the statement that the right way to study deep learning is by singularity theory, I would like to see either experiments or theory that give me insights about deep learning. This could be done by example by doing a theory on a toy model and showing that it holds for real deep learning applications (or at least for an array of synthetic distributions). I would like to qualify by saying that statistics is not my main field of study and I would be happy to receive clarifications if I misunderstood anything.

---

> ### Author Response · Authors · 2020-11-17
> **Response to AnonReviewer2**
>
> We are glad to hear that Reviewer 2 enjoyed the pedagogical aspect of our submission.
>
> Regarding the comments on the effective number of parameters. There are no assumptions on the model here: the calculation is meant only to exhibit that in regular models and a slightly larger class (what we term the minimally singular models) the RLCT agrees with a natural count of the effective number of parameters. The point of this calculation is to show that the RLCT is a generalisation of this quantity to general singular models (such as DNNs). While there may exist other generalisations, it is hard to argue they can be equally fundamental, given the role of the RLCT in the asymptotic formula for the Bayes generalisation error. We agree that these observations are elementary given a familiarity with singular learning theory, nonetheless they are important and have been missed.
>
> The claim that the Bayes predictive distribution is superior to MAP or MLE for singular  models is on firm theoretical grounds. This is hard to demonstrate empirically in a complete way because the posterior distribution over DNN weights is intractable. However, we were encouraged by the fact that being Bayesian in the last layer (which can be viewed as an approximate Bayes predictive distribution) shows superiority over MAP. We agree that more experiments should be done to support the claim that the last-layer-Bayesian approach is a good scheme for approximating the Bayes predictive distribution.

---

### Official Review · AnonReviewer1 · 2020-10-29
**insufficient intellectual contributions**

**Rating:** 4
**Confidence:** 5

**Review:**

This paper studies the manifold of the weights in a neural network. The paper discusses the singular learning theory approach of Sumio Watanabe and argues for more exploration of this theory for understanding generalization performance of deep networks.

My opinion of this paper is lukewarm. There is a large amount of existing work on singular learning theory. I agree with the paper that the approach is a promising direction to understand generalization in deep learning. However, the intellectual contributions of this particular paper to the existing literature are difficult to ascertain and insufficient warrant publication.

Some comments.
1. I like the section on how flatness of the energy landscape connects with Real log canonical threshold (RLCT). The RLCT gives a much more refined treatment of different measures of local geometry of the energy landscape used in the literature (Hessian-based, local-entropy based, Bayes free energy etc.).
2. The paper would benefit from Appendix A.1-4 to the main text. Some of these calculations have appeared elsewhere in the literature (see the work on K-FAC for the Fisher information) but the development in A.2-A.4 is interesting.

Some questions that the authors could think about.
1. The singular learning theory argues that the Bayes predictive distribution generalizes better MLE/MAP. How does this explain the fact that one sample from the posterior distribution also results in good generalization for deep networks?
2. Singular Learning Theory is not new. While it is true that this approach has been overlooked in the recent work on deep learning, better experimental evidence is necessary in order to make a convincing case that it is a promising one. The results shown here are accurate computations using MCMC for small models. Can you also investigate whether one obtains non-vacuous generalization bounds for large deep networks using an estimate for the RLCT?
3. Section 7: Is the true distribution of data really unrealizable? The reviewer is of the opinion that the fact that we can learn very good generative models for complex data indicates that it might be more viable to study the case when the true distribution is realizable for large-enough models.

---

> ### Author Response · Authors · 2020-11-17
> **Response to AnonReviewer1**
>
> We are glad the reviewer enjoyed the section on connections of RLCT with flatness. We agree that moving A.1-A.4 up to the main text would help the general exposition.
>
> Regarding the questions posed by the reviewer:
>
> “…the fact that one sample from the posterior distribution also results in good generalization for deep networks”
>
> This is a good question but we wish to reframe it slightly, since research on the connection between finding a solution by SGD and sampling from the posterior distribution is currently inconclusive. Regarding the question “why does one run of SGD tend to produce a network that generalises well” it is true that singular learning theory does not currently offer a conclusive answer. Progress will require a rigorous understanding of the connection between SGD and the posterior.
>
> However, this topic is addressed in Section 5 where the MAP is used a baseline to compare with our estimate of the Bayesian predictive distribution. The MAP is the solution reached by the SGD, and it is in general inferior to the approximate Bayesian predictive distribution.
>
> “…better experimental evidence is necessary in order to make a convincing case that it [singular learning theory] is a promising one”
>
> We are not aware of methods for estimating the RLCT for large DNNs. We hope to point out the importance of this quantity so that the plethora of talented people from the DL community can engage with this research. Also it is important to highlight the role of experiments in a work of this nature. In deep learning research where the theory is lacking, empirical evidence assumes a primary role. This is not the case here. We highlight a sound theoretical framework for studying deep learning but the challenge here is that many objects of interest (such as the RLCT) are difficult (but not impossible surely!) to estimate.
>
> “Can you also investigate whether one obtains non-vacuous generalization bounds for large deep networks using an estimate for the RLCT”
>
> The relationship between the RLCT and (average) generalization error of the Bayes predictive distribution is not in the form of a bound. What would be interesting is to explore bounds on the RLCT itself.
>
> “Is the true distribution of data really unrealizable? The reviewer is of the opinion that the fact that we can learn very good generative models for complex data indicates that it might be more viable to study the case when the true distribution is realizable for large-enough models.”
>
> The empirical success of deep generative models is mostly measured by its generative ability, e.g.. performance at drawing realistic looking images. The performance of deep generative models as density estimation is far less convincing. In fact, there are many emerging works that illustrate the phenomenon that a learned generative model can assign higher likelihood to Out-Of-Distribution instances than In-Distribution instances. This empirical observation casts serious doubt on whether generative models learn the underlying density well. It may learn it well enough to sample but that does not mean the generative model is estimating the density in a functionally accurate way.

---

### Official Review · AnonReviewer3 · 2020-10-29
**Useful probabilistic framework for singularity of NNs**

**Rating:** 5
**Confidence:** 3

**Review:**

The paper is a terse account of singularity of deep learning with a probabilistic view. Clearly written, gives an overview of the contributions and related work quickly and dives into the setup. The main byproduct of singularity is the inapplicability of classical methods. This is no news to many people in the field, yet I find the perspective provided in this work fresh and I think it has potential for further developments, although its current applicability is limited and it doesn’t say something that was not already known. Here are further comments:

- The paper would benefit a lot from clearly laying out the concepts and definitions. Especially section 3 would benefit a lot from such clarity and would help a wider audience to follow the work.
- The end of section 3 contains the key idea and would benefit from further clarity.
- How does the measure compare with the other frameworks? What does it imply for existing models (examples)?
- What is the relevance of the tasks and experiments at the end of section 5? How can one move from regression to high dimensional classification tasks?
- In light of the under/over-parametrization debate, can the framework account for such a phase transition, or can the experiments reflect this?
- What are the promises of this framework? What benefits would people in the community expect if they study this?

---

> ### Author Response · Authors · 2020-11-17
> **Response to AnonReviewer3**
>
> We attempted to give a self-contained rapid introduction to singular learning theory in Section 3. This was necessarily terse given the page limit. We do agree that the discussion at the end of Section 3 should’ve been further highlighted.
>
> Reviewer 1 asks how the measure compares to other frameworks. We assume the question is, specifically, how the RLCT compares to other complexity measures in deep learning. As far as we know the RLCT is the only complexity measure in deep learning with a strong theoretical basis. However the RLCT is hard to estimate, so it would certainly be interesting to investigate other proposed complexity measures to see if they can offer a cheap method of approximating the RLCT.
>
> There is no problem to move from building an approximate Bayesian predictive distribution for a regression task to a classification task. The experiments in Section 5 operate on the assumption that “being a little bit Bayesian (in the last layer of the network)” can already reap the benefits of being “fully Bayesian.” In particular, we hoped Section 5 would offer some preliminary empirical confirmation of the fact that Bayesian prediction is superior to MLE/MAP. Note however that the theory is definitive on this matter -- for singular models Bayesian prediction is superior to MLE/MAP -- and thus empirical evidence should be viewed as supplementary.
>
> It would be interesting to see if singular learning theory can explain the phase transition from under to over-parametrised. Our current experiments were not designed with this in mind. In general, variation of the number of parameters within a family of related model is not a well-studied topic within singular learning theory, and this is an interesting open problem.
>
> Regarding the promise of the framework, and what the benefits the community could expect. While we agree that many people know the classical methods fail, it is widely under-appreciated how profound the failure is and how difficult and deep the new ideas required to deal with this failure are. In our view any fundamental mathematical theory of deep learning must eventually grapple with these issues in some form, and singular learning theory is at present the only theory to do so.

---

### Decision · Program_Chairs · 2021-01-07
**Final Decision**

**Decision:**

Reject

**Comment:**

The paper proposes to introduce ideas from singular theory to deep learning. All reviewers agree that the work is not yet ready for publication. The key issue seems to boil down to the fact that the paper does not propose nor verify any clearly motivated scientific hypothesis. Relatedly, the work includes many too broad or unscientific claims such as "To understand why classical measures of capacity fail to say anything meaningful about DNNs". Such statements should be given more precisely and with a proper citation.

Based on this I have to recommend rejecting the paper. At the same time, I would like to thank the Authors for submitting the work for consideration to ICLR. I hope the feedback will be useful for improving the work.